# Thermal Analysis of THz Schottky Diode Chips with Single and Double-Row Anode Arrangement

**DOI:** 10.3390/mi15080959

**Published:** 2024-07-27

**Authors:** Zenghui Liu, Xiaobo Zhang, Zhiwen Liang, Fengge Wang, Yanyan Xu, Xien Yang, Xin Li, Yisheng Liang, Lizhang Lin, Xiaodong Li, Wenbo Zhao, Xin Cao, Xinqiang Wang, Baijun Zhang

**Affiliations:** 1State Key Laboratory of Optoelectronic Materials and Technologies, School of Electronics and Information Technology, Sun Yat-sen University, Guangzhou 510275, China; liuzh329@mail2.sysu.edu.cn (Z.L.); liangzhw29@mail2.sysu.edu.cn (Z.L.); wangfg@mail2.sysu.edu.cn (F.W.); xuyy87@mail2.sysu.edu.cn (Y.X.); yangxen@mail2.sysu.edu.cn (X.Y.); lixin293@mail2.sysu.edu.cn (X.L.); liangysh39@mail2.sysu.edu.cn (Y.L.); linlzh7@mail2.sysu.edu.cn (L.L.); lixd65@mail2.sysu.edu.cn (X.L.); zhaowb27@mail2.sysu.edu.cn (W.Z.); caox58@mail2.sysu.edu.cn (X.C.); 2State Key Laboratory for Mesoscopic Physics and Frontiers Science Center for Nano-Optoelectronics, School of Physics, Peking University, Beijing 100871, China; zhangxiaobomsc@163.com (X.Z.); wangshi@pku.edu.cn (X.W.)

**Keywords:** Schottky diodes, thermal characteristics, terahertz, anode arrangement

## Abstract

GaN Schottky diodes show great potential in high-power terahertz frequency multipliers. The thermal characteristics of GaN Schottky diodes with single and double-row anode arrangements are described in this paper. The temperature distribution inside the Schottky diode is discussed in detail under the coupling condition of Joule heat and solid heat transfer. The effects of different substrates and substrate geometric parameters on the thermal characteristics of the Schottky diode chips with single and double-row anode arrangements are systematically analyzed. Compared with that of the chip with single-row anode arrangement, the maximum temperature of the chip with double-row anode arrangement can be reduced by 40 K at the same conditions. For chips with different substrates, chips with diamond substrates can withstand greater power dissipation when reaching the same temperature. The simulation results are instructive for the design and optimization of Schottky diodes in the terahertz field.

## 1. Introduction

Terahertz waves have great potential applications, such as earth atmospheric remote sensing, biomedicine, high-speed communication, spectroscopic, and imaging techniques [1,2,3,4,5]. Terahertz sources are the key component for generating terahertz waves, which are essential for the development of terahertz electronic systems. At present, the frequency multiplier based on gallium nitride (GaN) planar Schottky barrier diode (SBD) has been developing rapidly and gaining great attention due to its high breakdown voltage, high electron mobility, and high thermal stability [6,7]. The current mainstream method to obtain a high output power is to withstand sufficient input power [8]. However, excessive input power generates an amount of heat that causes the temperature inside the diode to rise rapidly, and GaN SBD faces challenges related to the self-heating effect, which can degrade device performance and long-term reliability [9]. Therefore, chip thermal management has become an important aspect in the design of Schottky diode-based circuits for high-power applications [10,11,12,13,14].

Accurately predicting the anode junction temperature [15] and understanding the heat flow law inside the GaN SBD chips is the basis for developing effective thermal management strategies [16]. The heat transport process in GaN SBD is dominated by thermal conduction, with heat generated at the Schottky junction interface and transported through the GaN epitaxial layer, the GaN buffer layer, the substrate, and, finally, dissipated to the external environment [17]. In recent years, the thermal management of Schottky diode-based multipliers has been investigated in many research works. Aik Yean Tang et al. [18] presented a self-consistent electro–thermal model for multi-anode Schottky diode multiplier circuits. The thermal model is developed for an n-anode multiplier via a thermal resistance matrix approach. The model could predict the hot spot temperature in the frequency multiplier chip. Carlos et al. [19] presented the thermal analysis of different Schottky diode frequency doubler chip layouts. Cui et al. [20] presented the temperature distribution of the diode at dissipated powers, studied through electromagnetic heating multi-physics coupled with heat transfer in solids and electric currents. Song et al. [21] presented thermal characterization results of GaN Schottky diodes in the frequency multipliers with flip-chip configuration. These reports have provided valuable insights for our work with GaN SBD. However, reports on thermal analysis of the SBD chips with double-row anode arrangement by coupling Joule heat with solid heat transfer are limited.

In this work, a 3D thermal analysis of the planar Schottky diode chips with single and double-row anode arrangements is first systematically presented. The steady-state thermal characteristics of the chips are analyzed by solving the heat equation in the 3D calculation domain. This paper aims to gain insight into the internal heat distribution of Schottky diodes with different anode arrangements during operation. The heat distribution of Schottky diodes with single and double-row anode arrangements under the same excitation is studied, as well as the temperature distribution of Schottky diode chips with different substrate materials, including sapphire, silicon (Si), silicon carbide (SiC), and diamond. The effects of substrate geometrical parameters and anode spacing on the thermal characteristics of Schottky diode chips are systematically analyzed.

## 2. Thermal Simulation for Schottky Diode Chips

The 3D model of multi-anode planar Schottky diode chips is illustrated in Figure 1. As shown in Figure 1a,b, one chip comprises six anodes with single-row anode arrangement. The other comprises 12 anodes in an anti-series arrangement with double-row anode arrangement. Figure 1c shows the cross-view of the single-anode planar Schottky diode. The diode contains metal electrodes, a low-doped n^−^-GaN layer, a high-doped n^+^-GaN layer, a GaN buffer layer, and a substrate. Because the temperature distribution of the chip is affected by the area of the anode junction, the area of each anode junction on the single-row anode arrangement chip is 18.1 μm^2^. While the area of each anode junction on the double-row anode arrangement chip is 9.05 μm^2^, apart from this, their geometric and material parameters are the same.

In the simulation, the temperature distribution of the Schottky diode chip is analyzed by coupling Joule heating with solid heat transfer because the heat comes from the Joule heat generated by the current flowing through the Schottky diode in practice. In a frequency multiplier, the chip is connected to the waveguide block by two beam leads. So, the two facets of the substrate facing the waveguide are set as the ambient temperature (T_amb_), while other surfaces are assumed as adiabatic, i.e., the thermal radiation and the heat convection are ignored. For simplicity, the thermal properties of the epitaxial layer, the highly doped layer, and the buffer layer are assumed to be the same. In the simulation, the anode consists only of gold, although, in fact, the anode is composed of titanium (Ti), aluminum (Al), nickel (Ni), and gold (Au), as this has little effect on the thermal simulation results. The temperature-dependent thermal properties of semiconductors are considered. Table 1 lists the thermal conductivities and electric conductivities of the materials used in the simulations [22,23]. All calculations in this work are based on the finite-element method (FEM) as implemented in COMSOL multiphysics. Upon setting up the proper simulation boundaries and material properties, the temperature can be derived from the following formula (1):(1)$$Q=-\nabla \cdot k\left(T\right)\nabla T$$
where *T* is temperature, *k*(*T*) is the temperature-dependent thermal conductivity, and *Q* is the heat generation rate per unit volume.

When the diodes work, the heat comes from the Joule heat generated by the current flowing through the epitaxial layer. So, the temperature distribution of the Schottky diode chip is analyzed by coupling Joule heating with solid heat transfer. The current flowing through the Schottky diode can be obtained by surface integration of the current density on the cathode pad, and the thermal dissipation power P_diss_ generated by the Schottky diode during operation can be calculated by Joule’s law. Figure 2 shows the simulation results of the same device by using Joule heat and applying a heat source (P_diss_) to the anode junction. It can be seen that the overall temperature of the device calculated by coupling Joule heat with solid heat transfer is higher than that using only solid heat transfer. In theory, the coupling method of Joule heat and solid heat transfer is closer to the actual working state of the device than the method of only using solid heat transfer.

When the ambient temperature (T_amb_) is set as 293.15 K, the same voltage source is applied to the central pad of the two chips, respectively. The simulated overall temperature distribution of the chip with single-row arrangement is shown in Figure 3a. The simulated overall temperature distribution of the chip with double-row anode arrangement is shown in Figure 3b. When the Schottky diode is working, the current flowing through the diode generates a lot of heat at the anode junction, and the heat is transferred to the entire chip through thermal conduction. The maximum temperature of the chip with single-row anode arrangement is 31 K higher than that of the chip with double-row anode arrangement. In the chip, the anode junction near the center welded plate has the highest temperature. Figure 3c shows the heat flux around the anode. It can be found that heat flows around the anode junction, where the temperature is highest. So, the temperature distribution at the anode junctions of the chip was studied.

## 3. Results and Discussion

Figure 4a shows the temperature profile in the cross section centered on diode anodes for the SBD chips when the same voltage value is applied. The temperature at the anode of the chip with double-row anode arrangement is obviously lower than that of the chip with single-row anode arrangement. It is observed that the hot spot in the chip is located at the anode closest to the central pad, i.e., anode 1 (the chip with single-row anode arrangement) and anode 2 (the chip with double-row anode arrangement). This is because the heat is dissipated laterally through the beam leads, and the positions of anode 1 and anode 2 are further away from the side of the heat sink. In addition, anode 1 and anode 2 are more affected by thermal crosstalk than the other anodes. As Figure 4b shows, the temperature of every anode junction of the Schottky diode chip with double-row anode arrangement is lower than that of the Schottky diode chip with single-row anode arrangement. This is because the chip with double-row anode arrangement can transfer more heat to the substrate, relieving the heat gathered at the anode junction and reducing the anode temperature.

Figure 5a shows the temperature distribution of the yz-cross section centered on anode 1 and anode 2 of the single-row anode chip and the double-row anode chips with different anode spacing. It can be seen that the maximum temperature of the double-row anode arrangement chip is obviously lower than that of the single-row anode arrangement chip at the same dissipation power. For the chip with double-row anode arrangement, it is observed that the maximum temperature in the chip is reduced as the anode spacing increases. When the anode spacing is 28 μm, the maximum temperature of the double-row anode arrangement chip is reduced by 40 K compared with that of the single-row anode arrangement chip. Figure 5b shows the temperature distribution along the chip substrate surface through the anode center. It can be seen that the substrate temperature of the chip with double-row anode arrangement is lower in the anode junction region than that of the chip with a single-row anode arrangement, but the temperature of the other regions of the substrate is higher than that of the chip with single-row anode arrangements, which indicate that the substrate of the chip with double-anode arrangement absorbs more heat. As a result, the temperature at the anode junction of a chip with a double-anode arrangement is lower.

## 4. Effect of Substrate on Thermal Characteristics of SBD

From the material system point of view, gallium nitride (GaN) could be grown on silicon (Si) and silicon carbide (SiC) substrates. The material system of SiC and Si could possess a better heat handling capability than sapphire. So, the thermal characteristic of GaN-based diodes on Si and SiC-substrate is discussed. In addition, the substrate transfer technology has provided an opportunity to develop GaN-based diodes on a diamond substrate. That is, the non-substrate Schottky diode is bonded to the diamond substrate. Although diamond has a higher thermal conductivity than the substrate discussed earlier, this will produce a large thermal resistance at the bond, affecting the thermal characteristics of the Schottky diode. We assume that the increased thermal resistance due to the addition of the bonding layer is $R_{th}$. $R_{th}$ is related to the thickness and material of the bonding layer and the bonding technology. $R_{th}$ is between about 5 and 50 m^2^·K/GW [23,24,25]. In the simulation, $R_{th}$ is set as 10 m^2^·K/GW.

Figure 6 shows a comparison of the thermal behavior of the GaN-on-Si, SiC, and diamond substrate-based SBD chips at the same power dissipation. It can be observed that the maximum temperature of the chip decreases with the increase of the substrate thermal conductivity when the total dissipated power of the chips is the same. Since the Si has the lowest thermal conductivity in these materials, the hot spot temperature is the highest. Despite the influence of the bonding agent, the SBD chip on the diamond substrate still has the best thermal characteristics. In the case of the chip of single-row anode arrangement, the temperature rise values at hot spots of these chips are 237 K (Si), 142 K (SiC), and 123 K (diamond), respectively, while the temperature rise values at hot spots of these chips are 173 K (Si), 105 K (SiC), and 87 K (diamond) for the chip of double-row anode arrangement, respectively. No matter what kind of substrate the chip is on, the thermal characteristic of the SBD chip with double-row anode arrangement is better than that of the SBD chip with single-row anode arrangement. Thus, it is concluded that the thermal behavior of the SBD chip could be improved by bonding it to the diamond and utilizing double-row anode arrangement.

In the case of the SBD chip on the sapphire substrate, we have studied the influence of the geometric parameters of the substrate on the thermal characteristics of the chip with thermal simulations. Figure 7 and Figure 8 show the thermal characteristic of the chip on different thickness and width substrates, and it can be found that for the chips with the same substrate material and geometric parameters, the thermal characteristics of the chips with double-row anode arrangement are better than that of the chips with single-row anode arrangement. In addition, when the thickness and width of the sapphire substrate were increased, the maximum temperature of the chip decreased gradually, regardless of whether the chip has a single-row anode arrangement or double-row anode arrangement. Because a wider or thicker substrate has a higher heat capacity, this reduces the overall thermal resistance of the SBD chip. However, the temperature drop will be significantly mitigated when the thickness and width of the substrate are increased to 80 μm and 140 μm, respectively. For Schottky diodes operating under high-frequency conditions, excessive thickness and width not only lead to challenges in device assembly but also increase the loss of energy. Therefore, the self-heating effect of the chip could be weakened by appropriately increasing the width and thickness of the substrate.

We also analyze the steady-state maximum temperature in the SBD chip at different power dissipation levels, with the results shown in Figure 9. The stationary dissipated power is defined as a voltage–current multiplication. The current is obtained by integrating the current density on the cathode pad. It is observed that when the maximum temperature of the chip reaches about 550 K, the Schottky diode on the diamond could withstand higher power dissipation. At a lower dissipated power level, the anode junction temperature of the SBD chips with single and double-row anode arrangements is similar. However, it is observed that the maximum temperature between two chips deviates as the power dissipation level is increased. In addition, it can be found that for Schottky diodes on all substrates, when the maximum temperature reaches the same value, the chip with double-row anode arrangement has better thermal performance than the chip with single-row anode arrangement. According to the simulation results, when the dissipation power reaches about 1.7 W, the maximum temperature of the chip with a double-anode arrangement on the diamond substrate is only 469 K. Therefore, for the high-input power density operation, obvious improvement of thermal characteristics is obtained at the SBD chip with double-row anode arrangement and high thermal conductivity substrate.

## 5. Conclusions

In this paper, the thermal analysis of the chip with single and double-row anode arrangements was carried out with the coupling of Joule heat and solid heat transfer. The temperature-dependent thermal conductivity of the substrate and thermal resistance at the bonding layer were taken into account in the modeling. The thermal characteristics of Schottky diodes with single and double-row anode arrangements on different substrates are analyzed systematically. At the same dissipation power, the chip with double-anode arrangement has better thermal characteristics than the chip with single-anode arrangement. In addition, the self-heating effect of the Schottky diode chip can be further alleviated by using a substrate with higher thermal conductivity and appropriately increasing the geometric parameters of the substrate. It can be seen that the Schottky diode chip with double-row anode arrangement and a good thermal conductivity substrate is helpful in improving the performance of Schottky diode chips. In addition to the above conclusion, we can also conclude that the maximum temperature of the SBD chips with double-row anode arrangement on a diamond substrate is less than 500 K when the power dissipation is about 1.6 W. These results are instructive for the thermal management of Schottky diode chips in the high power density field.

## Figures and Tables

**Figure 1 micromachines-15-00959-f001:**
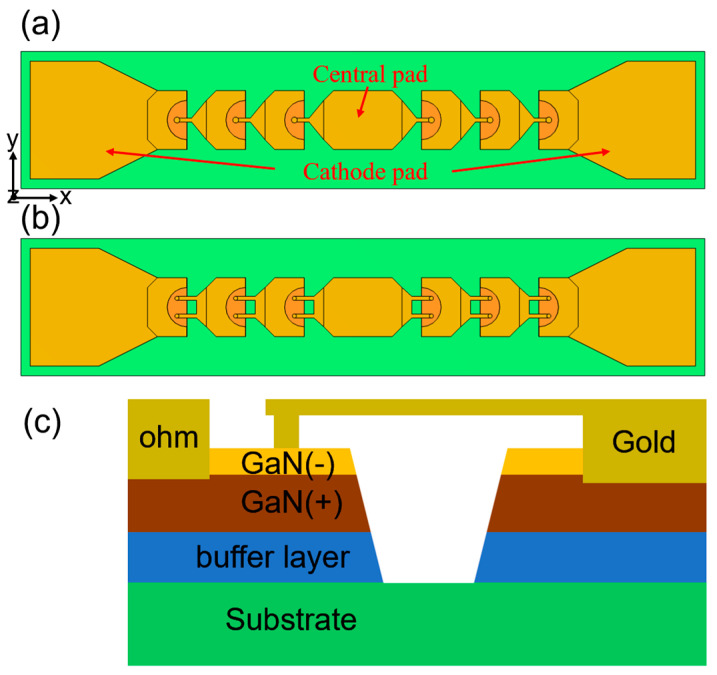
(**a**) The model of the Schottky diode chip with single-row anode arrangement. (**b**) The model of the Schottky diode chip with double-row anode arrangement. (**c**) Cross-view of a diode.

**Figure 2 micromachines-15-00959-f002:**
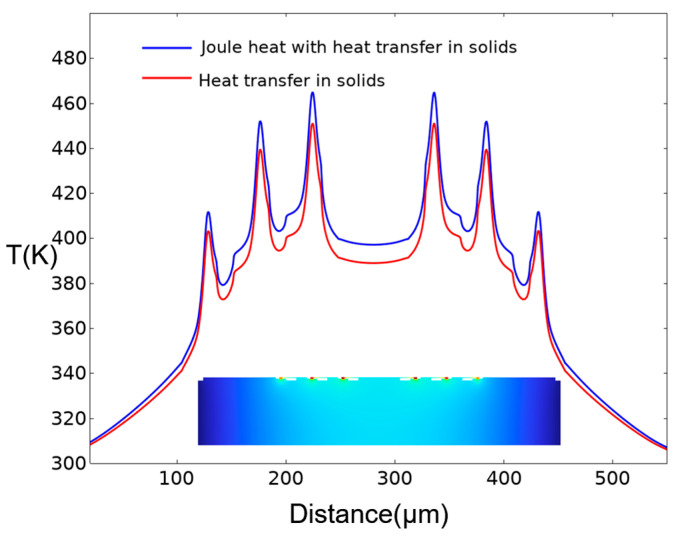
The temperature profile along the surface of the substrate of the Schottky diode chip by coupling Joule heating with solid heat transfer or only solid heat transfer. (Inset: Temperature distribution in the cross-sectional view of the chip).

**Figure 3 micromachines-15-00959-f003:**
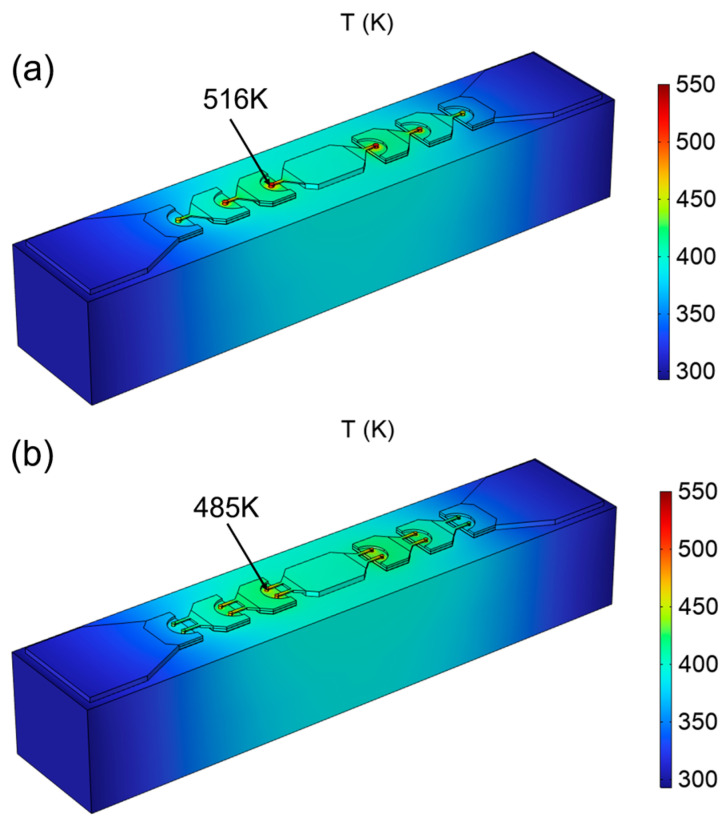
(**a**) The overall temperature distribution of the chip with single-row anode arrangement. (**b**) The overall temperature distribution of the chip with double-row anode arrangement. (**c**) Heat flux around the anode junction.

**Figure 4 micromachines-15-00959-f004:**
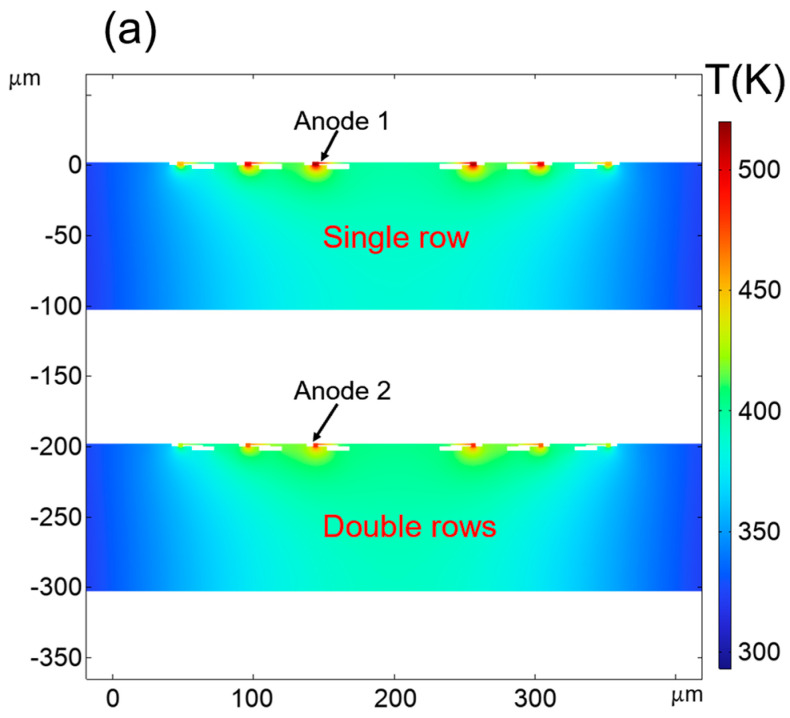
(**a**) Temperature profile in the cross-sectional view of the chip with single and double-row anode arrangements. (**b**) Temperature profile along the substrate surface and through the center of the anode.

**Figure 5 micromachines-15-00959-f005:**
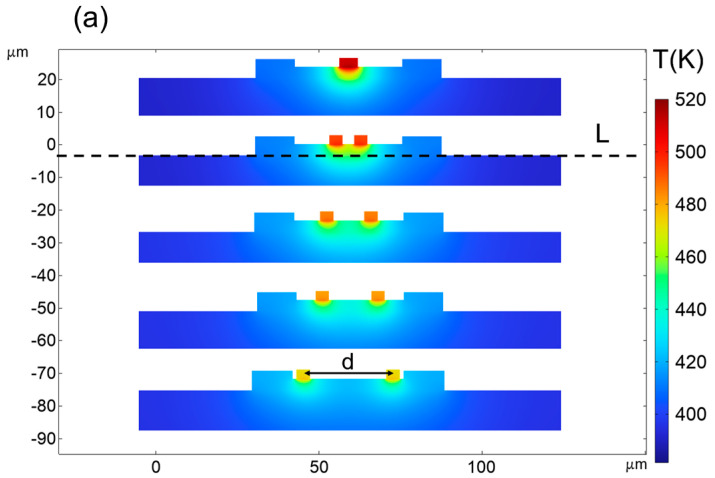
(**a**) Temperature distributions of the yz-cross section on anode 1 and anode 2. (**b**) Temperature profile along the substrate surface (line L).

**Figure 6 micromachines-15-00959-f006:**
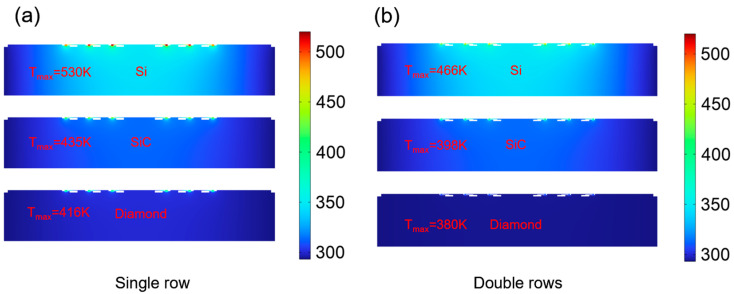
Cross-section temperature distribution along the center of the anodes on different su-strates. (**a**) The chip with single-row anode arrangement. (**b**) The chip with double-row anode a-rangement.

**Figure 7 micromachines-15-00959-f007:**
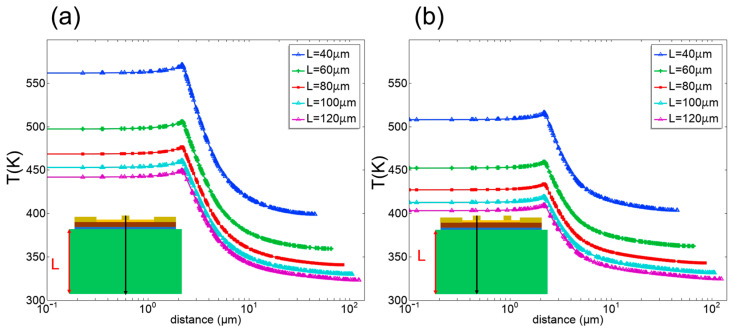
Temperature profile from the anode to the substrate bottom of the SBD chip with different thickness substrate. (**a**) The chip with single-row anode arrangement. (**b**) The chip with double-row anode arrangement.

**Figure 8 micromachines-15-00959-f008:**
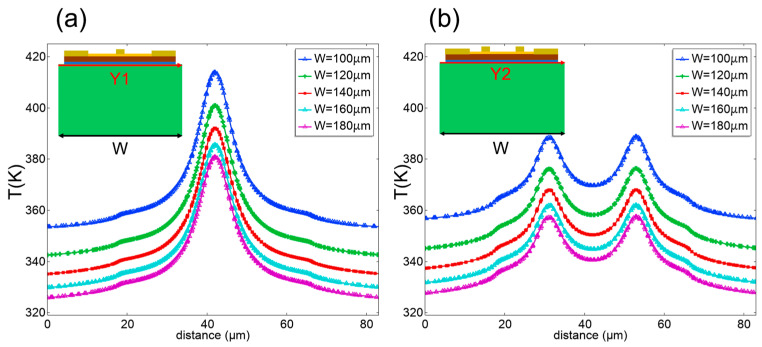
Temperature profile along the substrate surface and through the anode center of the SBD chip with different width substrate (line Y1 and line Y2). (**a**) The chip with single-row anode arrangement. (**b**) The chip with double-row anode arrangement.

**Figure 9 micromachines-15-00959-f009:**
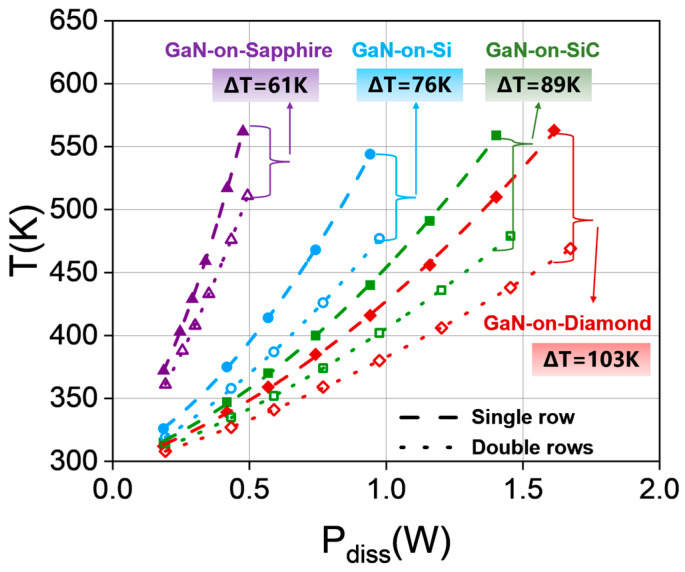
A comparison of the maximum temperatures in the chips of single and double-row anode arrangements on different substrates as a function of power dissipation level.

**Table 1 micromachines-15-00959-t001:** Material thermal parameters.

Material	Thermal Conductivity (W/mK)	Electric Conductivity (S/m)
Au	317	$4.56\times 10^{7}$
n^−^-GaN	157 × (300/T)^1.9^	560
n^+^-GaN	157 × (300/T)^1.9^	$1.6\times 10^{7}$
buffer layer	157 × (300/T)^1.9^	0.01
Si	160 × (300/T)^1.5^	-
sapphire	35 × (300/T)	-
SiC	400 × (300/T)	-
Diamond	1100	-

## Data Availability

The data that support the finding of this study are available from the corresponding author upon reasonable request.

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
