# Peer review of "Thermal Analysis of THz Schottky Diode Chips with Single and Double-Row Anode Arrangement"

_micromachines, 2024, doi:10.3390/mi15080959_

Round 1

Reviewer 1 Report

Comments and Suggestions for Authors

The manuscript is devoted to the thermal analysis of high-frequency Schottky barrier diode designs based on GaN.

1. GaN-based devices have been researched and developed for a long time. What is the fundamental novelty of your work? Increase your motivation.

2. The title of the manuscript indicates that the diodes are designed for the terahertz frequency range. What exactly makes them applicable to the terahertz frequency range? How do these diodes differ from microwave diodes based on GaN?

3. In my opinion, the attracted diode model (Figure 1) is primitive. Especially when considering the GaN-based HEMT designs and structures.

4. Imagine a cross section for the whole chip, not just a section.

5. Why were such a design and geometric parameters of the chips chosen?

6. Which program was used to simulate the temperature distribution and the volt-ampere characteristics? On what laws is the modelling based?

7. The conclusion that the diamond substrate is better than the others and the sapphire substrate is worse seems trivial. It is understandable even without modelling!

Author Response

Dear Referee:
The following is our reply according to the review comments and suggestions.

Comments and Suggestions for Authors
The manuscript is devoted to the thermal analysis of high-frequency Schottky barrier diode designs based on GaN.

1.GaN-based devices have been researched and developed for a long time. What is the fundamental novelty of your work? Increase your motivation.

Response: Thanks for the reviewer’s comments. The fundamental novelty of my work mainly includes the following two points:

  • In this paper, the maximum temperature and temperature distribution of Schottky diode chip are calculated by coupling Joule heat with solid heat transfer, which is closer to the state of Schottky diode when a single heat source is applied than in other literatures.
  • In this paper, the difference in thermal performance of the Schottky diode chips with single and double rows aondes arrangements on different substrates are analyzed.     

Response: Thanks for the reviewer’s suggestions. we have added the description for my motivation in the revised manuscript.

“These reports have provided valuable insights for our work with GaN SBD. However, reports on thermal analysis of the SBD chips with double anodes arrangement by coupling Joule heat with solid heat transfer are limited.”

2.The title of the manuscript indicates that the diodes are designed for the terahertz frequency range. What exactly makes them applicable to the terahertz frequency range? How do these diodes differ from microwave diodes based on GaN?

Response: Thanks for the reviewer’s comments. The parameters of the Schottky diode chain I simulated came from the Schottky diodes prepared in our laboratory.As shown in figure 1. We measured the capacitance-voltage (C-V) and current-voltage (I-V) characteristics of a diode by the Keysight B1505A. According to these characteristics, the series resistance(Rs) and zero-bias capacitance(Cj0) of Schottky diode can be calculated. Rs is 23.3Ω,Cj0 is 41.6fF. , so the Schottky diode can be operated in the terahertz band.

Response:The size of the terahertz Schottky diode is small, the series resistance(Rs) and junction capacitance(Cj0) compared to microwave devices are small. It can be operated in the terahertz band().

Figure 1. The capacitance-voltage (C-V) and current-voltage (I-V) characteristics of a diode.

3.In my opinion, the attracted diode model (Figure 1) is primitive. Especially when considering the GaN-based HEMT designs and structures.

Response: Thanks for the reviewer’s comments. According to some reports (a-d), I used the methods in these reports for modeling and simulation. Firstly, the model is constructed according to the actual object, and heat sources with a certain dissipation power is applied in the model, and reasonable boundary conditions are set according to the direction of heat transfer. Finally, the temperature distribution in the device is solved by Fourier heat transfer equation. In addition, because the thickness of gallium nitride layer(n--GaN layer, n+-GaN layer and GaN buffer layer) in the Schottky diode in this paper is more than 3, only Fourier heat transfer equation is used to solve the problem, ignoring the phonon ballistic-diffusive transport mentioned in report b.

(a) Dong, Y.; Liang, H.; Liang, S.; Zhou, H.; Yu, J.; Guo, H.; Zeng, H.; Feng, Z.; Zhang, Y. High-Efficiency GaN Frequency Doubler Based on Thermal Resistance Analysis for Continuous Wave Input. IEEE Trans. Electron Devices 2023, 70, 4565–4571, doi:10.1109/TED.2023.3294897.

(b) Gohel, K.; Zhou, L.; Mukhopadhyay, S.; Pasayat, S.S.; Gupta, C. Understanding of Multi-Way Heat Extraction Using Peripheral Diamond in AlGaN/GaN HEMT by Electrothermal Simulations. Semicond. Sci. Technol. 2024, doi:10.1088/1361-6641/ad4a66.

(c) Chen, X.; Tang, D. Thermal Simulations in GaN HEMTs Considering the Coupling Effects of Ballistic-Diffusive Transport and Thermal Spreading. IEEE Transactions on Components, Packaging and Manufacturing Technology 2023, 13, 1929–1943, doi:10.1109/TCPMT.2023.3331771.

(d) Song, X.; Zhang, L.; Liang, S.; Tan, X.; Zhang, Z.; Gao, N.; Zhang, Y.; Lv, Y.; Feng, Z. Thermal Analysis of GaN Schottky Diodes in the Terahertz Frequency Multipliers. In Proceedings of the 2019 IEEE International Conference on Electron Devices and Solid-State Circuits (EDSSC); IEEE: Xi’an, China, June 2019; pp. 1–2.

4.Imagine a cross section for the whole chip, not just a section.

Response: Thanks for the reviewer’s suggestions. We have changed Figure 6 to it . The cross-section temperature distribution of Schottky diode chips with different substrates along the center of the anode is shown.  

Figure 2. Cross-section temperature distribution along the center of the anodes on different substrates. (a)The chip with single row anode arrangement, (b) The chip with double rows anode arrangement.

  1. Why were such a design and geometric parameters of the chips chosen?

Response: Thanks for the reviewer’s comments. The geometric parameters of the chips in the simulation are chosen according to the actual size of the device. Figure 3 shows a picture of the device we made.

Figure 3. (a)The Schottky diodes chips. (b) Zoom-in view of the Schottky diodes chips.

  1. Which program was used to simulate the temperature distribution and the volt-ampere characteristics? On what laws is the modelling based?

Response: Thanks for the reviewer’s suggestions. We have added the description for it in the revised manuscript.

All calculations in this work are based on the finite-element method (FEM) as implemented in COMSOL Multiphysics. The laws that the modeling based are Joule's law and Fourier heat transfer equation.”

“When the diodes works, the heat comes from the Joule heat generated by the current flowing through the epitaxial layer. So the temperature distribution of the Schottky diode chip is analyzed by coupling Joule heating with solid heat transfer. The current flowing through the Schottky diode can be obtained by surface integration of the current density on the cathode pad, and the thermal dissipation power Pdiss generated by the Schottky diode during operation can be calculated by Joule's law. Figure 2 shows the simulation results of the same device by using Joule heat and applying heat source(Pdiss) to the anode junction. It can be seen that overall temperature of the device calculated by coupling Joule heat with solid heat transfer is higher than that by using only solid heat transfer. In theory, the coupling method of Joule heat and solid heat transfer is closer to the actual working state of the device than the method of only using solid heat transfer.”

  1. The conclusion that the diamond substrate is better than the others and the sapphire substrate is worse seems trivial. It is understandable even without modelling!

Response: Thanks for the reviewer’s comments. we have added the conclusion in the revised manuscript.

“In addition to the above conclusion, we can also concluded that the maximum temperature of the SBD chips with double rows anodes arrangement on diamond substrate is less than 500K when the power dissipation is about 1.6W. These results are instructive for the thermal management of Schottky diode chips in the high power density field.”

Reviewer 2 Report

Comments and Suggestions for Authors

This article reported simulation results from thermal analysis on THz Schottky barrier Diode (SBD) Chips with the single and double rows anode arrangement on different substrates. The results showed chip with double rows anode arrangement and a diamond substrate is beneficial to enhance the performance of SBD chips. Comments:

1.        Simulation results seem to be expected from chip with double rows anode arrangement and a diamond substrate, lacking of experimental results to prove this approach.

2.       Need to mention the software being used in the modeling simulation for SBD chips. However, this repots presented a good 3D thermal analysis on the planar Schottky diode chips.

Author Response

 Dear Referee:
The following is our reply according to the review comments.

Comments and Suggestions for Authors

This article reported simulation results from thermal analysis on THz Schottky barrier Diode (SBD) Chips with the single and double rows anode arrangement on different substrates. The results showed chip with double rows anode arrangement and a diamond substrate is beneficial to enhance the performance of SBD chips. Comments:

1.Simulation results seem to be expected from chip with double rows anode arrangement and a diamond substrate, lacking of experimental results to prove this approach.

Response: Thanks for the reviewer’s suggestions. Due to the small size of the device, our laboratory does not have an infrared imaging instrument with a resolution of micron level, so we cannot measure the actual working temperature of the device at present. When we contact a scientific research institution with a high-resolution infrared imager, we will verify these simulation results.

2.Need to mention the software being used in the modeling simulation for SBD chips. However, this repots presented a good 3D thermal analysis on the planar Schottky diode chips.

Response: Thanks for the reviewer’s suggestions. We have added the description for the software in the revised manuscript.

“All calculations in this work are based on the finite-element method (FEM) as implemented in COMSOL Multiphysics.”

Round 2

Reviewer 1 Report

Comments and Suggestions for Authors

The authors of the manuscript took into account the comments of the reviewer. The manuscript can be accepted for publication in the journal.

Reviewer 2 Report

Comments and Suggestions for Authors

This article can be published after the revision.